# Potential Circulating Biomarkers of Recurrence after Hepatic Resection or Liver Transplantation in Hepatocellular Carcinoma Patients

**DOI:** 10.3390/cancers12051275

**Published:** 2020-05-18

**Authors:** Dan G. Duda, Simona O. Dima, Dana Cucu, Andrei Sorop, Sebastian Klein, Marek Ancukiewicz, Shuji Kitahara, Speranta Iacob, Nicolae Bacalbasa, Dana Tomescu, Vlad Herlea, Cristiana Tanase, Adina Croitoru, Irinel Popescu

**Affiliations:** 1Steele Laboratories for Tumor Biology, Department of Radiation Oncology, Massachusetts General Hospital and Harvard Medical School, 100 Blossom St., Cox-734, Boston, MA 02114, USA; skitahara@twmu.ac.jp; 2Center of Digestive Diseases and Liver Transplantation, Fundeni Clinical Institute, 022328 Bucharest, Romania; dima.simona@gmail.com (S.O.D.); msiacob@gmail.com (S.I.); nicolaebacalbasa@gmail.com (N.B.); 3Center of Excellence in Translational Medicine, Fundeni Clinical Institute, 022328 Bucharest, Romania; sorop_andrei@yahoo.com (A.S.); danatomescu@gmail.com (D.T.); herlea2002@yahoo.com (V.H.); adina.croitoru09@yahoo.com (A.C.); 4Department of Anatomy, Animal Physiology and Biophysics, Faculty of Biology, University of Bucharest, 030018 Bucharest, Romania; dana.cucu@bio.unibuc.com; 5Institute for Pathology, University Hospital Cologne, 50937 Cologne, Germany; sebastian.klein@uk-koeln.de; 6Probabilitas LLC, Acton, MA 01720, USA; ma0089@gmail.com; 7Department of Anatomy, Tokyo Women Medical University, Tokyo 162-8666, Japan; 8Department of Anesthesia and Intensive Care, Fundeni Clinical Institute, 022328 Bucharest, Romania; 9Department of Pathology, Fundeni Clinical Institute, 022328 Bucharest, Romania; 10Department of Biochemistry-Proteomics, Victor Babes National Institute of Pathology, 050096 Bucharest, Romania; cristianatp@yahoo.com; 11Department of Medical Oncology, Fundeni Clinical Institute, 022328 Bucharest, Romania

**Keywords:** liver resection, transplantation, HCC, biomarkers, cytokine

## Abstract

*Background:* Improving surgical outcomes in hepatocellular carcinoma (HCC) patients would greatly benefit from biomarkers. Angiogenesis and inflammation are hallmarks of HCC progression and therapeutic targets. *Methods:* We retrospectively evaluated preoperative clinical variables and circulating (plasma) biomarkers of angiogenesis and inflammation in a cohort of HCC patients who underwent liver resection (LR) or transplantation (LT). Biomarker correlation with outcomes—freedom of liver recurrence (FLR), disease-free survival (DFS) and overall survival (OS)—was tested using univariate and multivariate Cox regression analyses. *Results:* Survival outcomes associated with sVEGFR1, VEGF and VEGF-C in LT patients and with IL-10 in LR patients. Moreover, in LT patients within Milan criteria, higher plasma VEGF and sVEGFR1 were associated with worse outcomes, while in those outside Milan criteria lower plasma VEGF-C associated with better outcomes. Multivariate analysis indicated that adding plasma VEGF or VEGF-C to a predictive model including Milan criteria and AFP improved prediction of DFS and OS (all *p* < 0.05). *Conclusion:* Survival outcomes after LR or LT differentially associated with angiogenic and inflammatory biomarkers. High plasma VEGF correlated with poorer prognosis within Milan criteria while low plasma VEGF-C associated with better prognosis outside Milan criteria. These candidate biomarkers should be further validated to improve patient stratification.

## 1. Introduction

Hepatocellular carcinoma (HCC) is the fifth most common malignant tumor and a leading cause of cancer-related death worldwide [1]. Curative treatments for HCC include surgical interventions: liver resection (LR) and liver transplantation (LT). However, the rate of local recurrence and extrahepatic metastasis following these surgical interventions remains high, which accounts in part for the high mortality from HCC [2]. Clearly, further improving outcomes in HCC using LR or LT will depend on optimal patient selection for these interventions.

Currently, treatment selection is based on staging criteria such as the Barcelona clinical liver cancer (BCLC) score, which stratifies HCC patients into five stages of the disease [3]. The model for end-stage liver disease (MELD) score (with higher scores indicating worse liver function) is associated with liver failure and mortality after LR for HCC [4,5]. In addition, Milan criteria is widely used for selection of HCC patients that are candidates for LT (i.e., radiologic tumor size and number: solitary tumor ≤5 cm or ≤3 tumors each <3 cm) [6]. However, multiple studies have shown that some patients within Milan criteria may have a poor outcome. Specifically, despite these strict criteria, the 5-year recurrence rate after LT remains approximately 10–15% for HCC.

Conversely, other patients who are beyond Milan criteria can have favorable outcomes post-LT. For example, a prospective study using BCLC-expanded criteria indicated that proper selection of candidates for extended indications of living donor LT for HCC patients may provide survival outcomes comparable to those obtained within the Milan criteria [7]. In addition, other studies have proposed expanded selection criteria, some of which included blood circulating biomarkers, to improve patient selection and treatment [8]. For example, a recent score incorporated circulating alphafetoprotein (AFP) level in the selection algorithm for surgical HCC patients [9,10]. Inclusion of additional circulating biomarkers could potentially improve prognostication, but also provide a better understanding of the determinants of disease recurrence and thus help guide the development of future combination treatments.

Progression from liver damage (inflammation and cirrhosis) to HCC is linked to pathological new blood vessel formation, largely driven by vascular endothelial growth factor (VEGF) receptor 2 (VEGFR2) pathway activation in endothelial cells, which disrupts the liver vascular architecture and induces the formation of portal-systemic collaterals and sinusoidal capillarization. High circulating levels of VEGF associate with a poorer prognosis in HCC. For example, high pre-operative values of plasma VEGF associated with shorter overall survival (OS) and disease-free survival (DFS) in HCC patients [11]. Moreover, inhibitors of VEGFR2 pathway, such as sorafenib, regorafenib, lenvatinib, cabozantinib or ramucirumab, have increased median OS in randomized phase III trials in patients with advanced HCC [12,13,14,15,16]. Circulating levels of pro-angiogenic (VEGF, PlGF) factors have been extensively tested as biomarkers of response in advanced disease [17]. Finally, the naturally occurring form of soluble (s)VEGFR1 (or sFLT1) is an endogenous inhibitor of the VEGF pathway, which has been also been associated with response to antiangiogenic therapies in HCC and other cancers, but also with underlying liver disease [17,18,19,20].

Another VEGFR2 ligand is VEGF-C, which is also a ligand for VEGFR3 and has been primarily linked with lymphangiogenesis. However, both VEGFR2 and VEGFR3 are expressed on normal liver and tumor endothelial cells, indicating that VEGF-C may potentially be a relevant proangiogenic factor in HCC [21,22]. Moreover, intratumoral VEGF-C expression correlated with HCC progression in HCC patients after LR [23]. Interestingly, circulating VEGF-C levels were associated with increased tumor necrosis after transarterial chemoembolization (TACE) prior to LT [24].

In addition to pathological angiogenesis, progression from liver damage to HCC is also linked to pathological inflammatory responses mediated by increased levels of cytokines such as tumor necrosis factor (TNF)-α, interleukin (IL)-1β, IL-6, IL-8 and IL-10. These cytokines play pleiotropic roles in liver inflammation and immunosuppression in tumors, but also angiogenesis and fibrosis; thus, they could promote HCC progression and treatment resistance, particularly in patients with underlying liver disease [20,25,26,27,28,29,30].

Here, we evaluated the association between survival outcomes and circulating angiogenic and inflammatory biomarker levels in blood samples prospectively collected in an observational study (HEPMARK) in HCC patients who underwent surgical treatments at one institution (Fundeni Clinical Institute, Bucharest, Romania).

## 2. Results

### 2.1. Patient Characteristics

The study included 180 patients (134 male and 46 female) diagnosed with HCC who underwent surgical interventions. These patients were eligible, signed informed consent and had blood samples banked between 2003 and 2016. All patients were treated at the Center for Digestive Disease and Liver Transplantation of Fundeni Clinical Institute, Bucharest, Romania. Patient characteristics are listed in Table 1. Among these 180 patients, 120 patients underwent LR (67%), defined as complete removal of their tumors with pathologically negative margins and 60 patients underwent LT (33%). Mean age was 64 years for LR patients versus 57 years for LT patients (*p* < 0.0001, see Table 1).

HBV infection was the most common risk factor in these HCC patients and was present in a higher proportion of patients in LT cohort (70.0%) than in LR cohort (41.2%) (*p* = 0.003). In addition, the vast majority of patients who underwent LT had liver cirrhosis (59/60 or 98.3%). Most of the patients in LR group had well preserved liver function, with Child–Turcotte–Pugh (CTP) scores A in 65% cases (78/120) and B in 5.8% cases (7/120). Among the patients in LT group, 71.2% (42/59) cases were CTP score B, while 28.8% (17/59) were CTP score A. The differences in liver function scored by CTP class were statistically significant between LT and LR groups, *p* < 0.0001. Moreover, the median value of MELD score was 8 (range, 6–11) in patients from LR group, and 13 (range 7–28) in those from LT group, a difference which was significant, *p* < 0.0001. A total of 33/51 (64.7%) patients evaluated for LT were considered to be within Milan criteria and 28/60 (47%) received a bridging therapy, most frequently using TACE (26/60 or 43%).

### 2.2. Post-Surgical Outcomes and Treatment

During the median follow-up time of 52.8 months [95% CIs: 41.6, 73.0 months] (accounting for mortality by inverse censoring method), 95/180 (53%) patients had died—74/120 (61.6%) in LR group and 21/60 (35.0%) in LT group. Moreover, 11 of the surviving patients developed recurrent disease (6.1%). A total of 54/180 patients (30%) had liver recurrence—46 (38.3%) in LR group and 8 (13.3%) in LT group. The median duration of freedom of liver recurrence (FLR) was +∞ months [56.8, +∞] months; median FLR was 92.5 [21.2, +∞] months for LR and +∞ [76.2, +∞] months for LT patients (*p* = 0.0014) (Figure 1A). Median DFS was 28.4 [95% CIs: 19.1, 52.8] months in the whole cohort; Median DFS was 19.6 [12.1, 29.1] months for LR and +∞ [39.0, +∞] months for LT patients (*p* = 0.0017) (Figure 1B). Median OS was 48.4 [32.3, 67.3] months in the whole cohort; median OS was 33.7 [26.8, 56.4] months for LR and +∞ [39.0, +∞] months for LT patients (*p* = 0.056) (Figure 1C).

### 2.3. Levels of Circulating Inflammatory and Angiogenic Biomarkers in Surgical HCC Patients

Table 2 summarizes the baseline levels of inflammatory and angiogenic biomarkers measured in plasma samples available from 45 of the HCC patients who underwent LR and all 60 HCC patients who underwent LT. Overall, there were significantly higher concentrations of plasma PlGF and lower concentrations of plasma IL-6, IL-8, bFGF, sVEGFR1, sTIE2 and VEGF-D in patients who underwent LR compared to LT patients. There were no significant differences in the plasma levels of IFN-γ, IL-10, TNF-α, VEGF, VEGF-C or serum AFP between the two cohorts. Serum samples were available from 69 of the HCC patients who underwent LR. When comparing plasma versus serum biomarkers among LR patients, we detected significantly higher levels of IL-8, bFGF, VEGF, VEGF-C, and VEGF-D and lower levels of sVEGFR1 and sTIE2 in serum (Appendix A). Therefore, we report here only the biomarkers measured in plasma for the correlative analyses in this study.

### 2.4. Correlation between Clinical Variables with Recurrence and Survival after Surgical Treatments

We next examined the association between background clinical variables used as prognostic markers—CTP score, Milan criteria, Edmondson–Steiner grade, tumor size, viral infection—and outcomes after surgical treatments (Table 3). In the overall cohort, OS and DFS were associated with Milan criteria (HR = 0.53, *p* = 0.0037 and HR = 0.64, *p* = 0.034, respectively) and with tumor size (HR = 1.06, *p* = 0.042 and HR = 1.06, *p* = 0.050, respectively); FLR was associated with HBV risk factor (HR = 2.08, *p* = 0.0097) and showed statistically insignificant trends for association with Milan criteria (HR = 0.58, *p* = 0.065) and Edmondson–Steiner grade (HR = 4.45, *p* = 0.054).

When evaluated after stratification by surgical intervention, OS in LR patients was associated with CTP score (HR = 3.26, *p* = 0.0006) and Milan criteria (HR = 0.53, *p* = 0.011), DFS was associated with CTP score (HR = 2.71, *p* = 0.0029) and FLR correlated with CTP score (HR = 2.72, *p* = 0.039) and HBV infection (HR = 2.14, *p* = 0.011). In addition, FLR showed statistically insignificant associations with Edmondson–Steiner grade (HR = 3.79, *p* = 0.086) and tumor size (HR = 1.02, *p* = 0.067). None of these clinical variables associated with outcomes in LT group, although both DFS and FLR showed a tendency for association with tumor size (HR = 1.27, *p* = 0.089 and HR = 1.54, *p* = 0.067, respectively). Of note, there was no significant correlation between any of the plasma biomarkers and prior TACE treatment in these patients.

### 2.5. Correlation between Circulating Inflammatory and Angiogenic Factors with Recurrence and Survival after Surgical Treatments

We next determined the association between plasma biomarkers and OS or risk of HCC recurrence (Table 4). In the overall cohort, plasma VEGF was associated with DFS (HR = 1.25, *p* = 0.019) and VEGF-C showed a trend for correlation with OS (HR = 1.67; *p* = 0.058) and DFS (HR = 1.49; *p* = 0.071) (Table 4).

However, when stratified by treatment, plasma VEGF was significantly correlated with DFS (HR = 1.38, *p* = 0.012) and OS (HR = 1.31, *p* = 0.037) and showed a tendency for association with FLR (HR = 1.51, *p* = 0.073) and plasma VEGF-C was also significantly associated with DFS (HR = 2.47, *p* = 0.033) and OS (HR = 2.49, *p* = 0.039), in LT patients. Moreover, plasma sVEGFR1 was associated with OS in LT patients (HR = 0.74, *p* = 0.045), but showed an inverse trend for correlation with OS in LR patients (HR = 1.38, *p* = 0.061).

Of immune cytokines, in LR group, plasma IL-10 and IL-6 were significantly correlated with OS (HR = 1.79, *p* = 0.0008; and HR = 1.34, *p* = 0.04, respectively); IL-10 and IL-8 were significantly correlated with DFS (HR = 1.35, *p* = 0.040 and HR = 1.54, *p* = 0.0093, respectively); and IL-8 was significantly correlated with FLR (HR = 1.63, *p* = 0.018). In LT group, plasma IFN-γ was correlated with OS (HR = 0.61; *p* = 0.043) and showed a statistically insignificant trend for association with DFS (HR = 0.66; *p* = 0.071) (Table 4).

### 2.6. Elevated Gene Expression Levels of IL10 Downstream Mediators IL10RA and JAK1 Are Associated with Worse OS in HCC

To further examine the role of IL-10, we evaluated whether mediators IL-10 pathway—receptors or downstream factors activated by IL-10—are also correlated with worse OS in another cohort of LR patients. To this end, we first investigated both *IL10RA* and *IL10RB* and its downstream targets *JAK1* and *TYK2* gene expression levels in 370 HCC tumor samples using the publicly available The Cancer Genome Atlas (TCGA) dataset. Both *IL10RA* and *JAK1*, as well as *IL10RB* and *TYK2* expression levels were clustered (Appendix A). *TYK2* did not show a positive correlation with neither *IL10RA* or *IL10RA* nor did *Il10RB* with *JAK1* expression. However, we found a significant correlation between *IL10RA* and *JAK1* expression levels (Appendix A). Furthermore, we explored whether these IL-10 pathway mediators were associated with OS in this HCC cohort. We found that patients whose tumors had high expression levels of *Il10RA* and *JAK1* (i.e., a z-score of more than 2 for mRNA expression) had a significantly shorter median OS of 37.29 months, compared to patients whose tumors showed low expression of either *Il10RA* and *JAK1* genes—median OS of 58.84 months (*p* = 0.021, chi-squared test) (Appendix A).

### 2.7. Correlation between Circulating Biomarkers and Outcomes after Stratifying by Surgical Intervention and Milan Criteria

We next examined the correlations between plasma biomarkers and outcomes when HCC patients were further stratified by surgical intervention type and Milan criteria score.

For LT patients within Milan criteria, plasma VEGF was correlated with DFS (HR = 1.45 and *p* = 0.044) and OS (HR = 1.44, *p* = 0.031) and plasma sVEGFR1 was associated with both DFS (HR = 0.64, *p* = 0.044) and OS (HR = 0.63, *p* = 0.042) (Table 5). For LR patients within Milan criteria, plasma sVEGFR1 showed an opposite correlation with FLR (HR = 3.08, *p* = 0.022) and DFS (HR = 2.40, *p* = 0.028) (Table 5). None of the inflammatory cytokines measured associated with outcomes in these subgroups of patients (Table 5).

For LT patients outside Milan criteria, only plasma VEGF-C was associated with DFS (HR = 6.15, *p* = 0.0082) and OS (HR = 5.41, *p* = 0.014) (Table 6). For LR patients outside Milan criteria, only plasma IL-10 was associated with OS (HR = 1.56, *p* = 0.021) (Table 6); of note, plasma IL-10 showed a statistically insignificant tendency for association with DFS (HR = 1.38, *p* = 0.060), plasma sVEGFR1 showed tendency for correlation with OS (HR = 1.43, *p* = 0.073) and plasma IL-8 with FLR and DFS (HR = 1.59, *p* = 0.093 and HR = 1.53, *p* = 0.054, respectively) (Table 6).

### 2.8. Analysis of Clinical Outcomes in a Proportional Hazards Model Including AFP and Plasma VEGF or VEGF-C after Adjusting for Milan Criteria Score

Given the association detected for plasma VEGF and VEGF-C in LT group, we next performed a further analysis in the proportional hazards (PH) regression model. We used log-transformed plasma VEGF or VEGF-C and AFP measurements and either: (1) stratified the patients based on Milan criteria, with likelihood ratio test performed to compare the model including only Milan criteria (as a stratum variable) and AFP, with a full model including Milan criteria, AFP and VEGF-C; or (2) included Milan criteria in regression, with likelihood ratio test performed to compare the model including only Milan criteria and AFP, with a full model including Milan criteria, AFP and VEGF-C. We found no difference in OS, DFS or plasma VEGF, VEGF-C or AFP biomarker levels between HCC patients within versus those outside Milan criteria (Appendix A). C-statistic values for PH model for OS and DFS with AFP, VEGF and Milan (as a stratum) were c = 0.674 and c = 0.690, respectively; for OS and DFS with AFP, VEGF and Milan (in regression) were c = 0.700 and c = 0.729, respectively (Table 7). The likelihood ratio test for adding VEGF to a model including only Milan criteria (as a stratum) and AFP yielded a χ^2^ of 6.12 for OS and of 8.22 for DFS, with 1 degree of freedom, *p* = 0.013 and *p* = 0.0042, respectively. The likelihood ratio test for adding VEGF to a model including only Milan criteria (in regression) and AFP yielded a χ^2^ of 5.98 for OS and of 8.44 for DFS, with 1 degree of freedom, *p* = 0.015 and *p* = 0.0037, respectively.

Moreover, C-statistic values for PH model for OS and DFS with AFP, VEGF-C and Milan (as a stratum) were c = 0.629 and c = 0.620, respectively; for OS and DFS with AFP, VEGF-C and Milan (in regression) were c = 0.686 and c = 0.692, respectively (Table 7). The likelihood ratio test for adding VEGF-C to a model including only Milan criteria (as a stratum) and AFP yielded a χ^2^ of 5.86 for OS and of 6.18 for DFS, with 1 degree of freedom, *p* = 0.016 and *p* = 0.013, respectively. The likelihood ratio test for adding VEGF-C to a model including only Milan criteria (in regression) and AFP yielded a χ^2^ of 6.54 for OS and of 5.95 for DFS, with 1 degree of freedom, *p* = 0.011 and *p* = 0.015, respectively.

These analyses suggest that adding plasma VEGF or VEGF-C to a predictive model including only Milan criteria and AFP could improve prediction of both OS and DFS.

## 3. Discussion

New strategies to select HCC patients for surgical interventions by incorporating biomarkers remain highly desirable. Starting with the adoption of Milan criteria in 1996 [6], several selection criteria, including tumor radiologic and pathologic characteristics (number, diameter), circulating biomarkers (serum AFP) and dynamic response evaluation of neoadjuvant therapies (e.g., TACE) have been developed to improve the prediction of recurrence after LT in HCC patients [31]. However, Milan criteria are restrictive and based on pre-LT tumor parameters and not on tumor biology of the explant [32]. Moreover, even for patients with HCC beyond the Milan criteria, the risk of recurrence may be as high as 70% at 2 years after LT [33].

To address this limitation, additional blood circulating biomarkers could provide a quantitative tool for HCC prognosis or prediction of recurrence risk. Due to their minimally invasive, objective, and reproducible characteristics, circulating biomarkers may be easily implemented for an unbiased estimation to complement existing scoring systems, which are largely based on clinical variables. However, the approaches and the impact of incorporating circulating biomarkers as a selection criterion for LR or an exclusion criterion for LT remain unclear. Incorporating one or multiple biomarkers into the Milan criteria could identify expanded criteria with low risk of recurrence after LT. One important goal is selection of patients using circulating biomarkers among HCC patients beyond Milan criteria who could achieve after LT comparable outcomes to patients within Milan criteria. Another important goal is to identify HCC patients within Milan criteria with aggressive tumors who may not derive a benefit from LT.

Here, we report significant associations for plasma levels of VEGF family members of pro-angiogenic proteins and post-surgical outcomes. VEGF family is critical in the pathogenesis of liver diseases, including in cirrhosis and hepatocarcinogenesis, through regulation of angiogenesis [25,34]. VEGF pathway is also an important target for therapy [17]. VEGF-C and VEGF-D are considered key lymphangiogenic factors, which may facilitate metastasis to lymph nodes. Of note, tumor size, Milan criteria score (which is heavily weighted by tumor size) and HBV etiology were some of the only clinical variables that associated with survival outcomes in our study.

Several reports have shown that pre-LT circulating AFP could be a useful predictor of the HCC recurrence, when using certain cutoff values [35,36,37,38]. The AFP model, which combined AFP serum values at listing with the usual criteria of tumor size and number has shown that AFP level at listing was an independent predictor of recurrence after LT for HCC and also predicted survival. The AFP model identified a subset of HCC patients exceeding Milan criteria, with AFP levels of <100 ng/mL at low risk of recurrence and with 5-year survival rates close to 70% and a subset of patients within Milan criteria, with serum AFP values greater than 1000 ng/mL at high risk of recurrence and significantly shorter survival [36]. As a result, circulating AFP is now recommended by some guidelines for surveillance of HCC patients with high risk of recurrence. However, the American Association for the Study of Liver Diseases (AASLD) excluded AFP from surveillance in 2010 and the EASL did not recommend it as a specific diagnostic test [39]. Therefore, additional circulating biomarkers are urgently needed for assessing the risk of HCC recurrence.

Importantly, our study found differential correlations between plasma sVEGFR1 and outcomes after LR versus LT. Interestingly, high sVEGFR1 levels associated with longer OS in LT patients, but the opposite tendency was seen in LR patients. Moreover, after stratifying the patients by using Milan criteria, we found that high sVEGFR1 levels in patients within the Milan criteria associated significantly with poorer outcomes in LR patients, but superior DFS and OS in LT patients. The benefit seen with high sVEGFR1 may be due to systemic inhibition of VEGF pathway [40] in cirrhotic patients in LT group [18]. In line with this hypothesis, plasma sVEGFR1 was significantly associated with MELD score in LT patients (*p* < 0.0001).

In addition, plasma VEGF and VEGF-C levels associated with poor outcomes in LT patients but showed no association in LR patients. Our data are consistent with previous reports which showed the role of serum VEGF in evaluating the risk of recurrence after surgical treatment in HCC patients [41]. Our analysis further indicates that high VEGF and low sVEGFR1 in plasma (i.e., surrogate biomarkers of active angiogenesis) may be poor prognostic factors for patients inside Milan criteria while low plasma VEGF-C (a surrogate biomarker of lymphangiogenesis and angiogenesis) may be a good prognostic factor in those outside Milan criteria. These results indicate a potential differential biomarker value—and potentially distinct biologic effect—of VEGF family proteins in disease recurrence depending on the severity of liver damage. Of note, we calculated C-statistics for models involving plasma VEGF and VEGF-C and detected the best values when we included Milan criteria scoring in the model as a regression variable, rather than stratifying on it.

Another group of circulating factors intimately involved in the progression of both cirrhosis and hepatocarcinogenesis consists of inflammatory cytokines. Previous studies reported the biomarker potential of circulating serum levels of key cytokines such as IL-10 and IL-6. One study reported that serum IL-10 and IL-6 levels correlated with tumor size in HCC patients. In another study, Yi Ren et al. observed that high IL-8 levels associated with larger tumors (>5 cm), advanced disease and tumor progression in HCC patients who underwent resection [42]. In addition, preoperative serum samples from 60 patients with resectable HCC showed that high IL-10 levels associated with shorter DFS [43]. Finally, circulating IL-8 was proposed as a prognostic biomarker in HCC [42]. In our study, we found increased IL-6 and IL-8 levels in the plasma of LT patients, which included a higher proportion of cirrhotic patients versus LR patients. Interestingly, significant correlations between plasma IL-6, IL-8 and IL-10 levels and survival outcomes were seen in LR patients, but not in LT patients. These correlations appeared to be driven by more advanced disease stage, i.e., in LR patients outside Milan criteria. Importantly, inflammation is known to be intimately involved in the progression of both liver cirrhosis and carcinogenesis (e.g., IL-10 [43]). To further examine the role of IL-10 pathway, we explored the association between genes downstream IL-10 signaling and OS in a cohort of HCC patients from the TCGA. In line with our circulating biomarker data, we found that a high level of *IL10RA* or *JAK1* gene expression in tumor tissue was significantly associated with shorter survival after LR in the HCC patients from the TCGA dataset.

On the other hand, plasma high IFN-γ levels associated with superior OS (and tended to correlate with longer DFS) in LT patients, which supports the notion that immune responses are important in controlling disease progression after LT.

Finally, most of the previous biomarker studies have examined serum protein levels in HCC patients. In our studies, we detected significant differences between plasma and serum biomarker levels, likely owing to the release of these proteins from platelet granules during serum sample preparation. More important, the associations seen between plasma biomarkers and outcomes were not seen when testing serum biomarkers. Thus, caution is recommended when interpreting and comparing data from serum versus plasma biomarker studies.

Our study has several limitations, including the limited sample size and single institutional analysis. In addition, future studies should investigate the roles of blood versus lymphatic vessel formation in HCC recurrence (hematogenous versus lymphatic metastasis) after LT. The C-statistic values reported here for VEGF and VEGF-C models are inferior to the one reported for OS in the Metroticket 2.0 publication (0.78) [31]. It is important to note, however, that Metroticket study endpoint was HCC-specific death while the one in this study was death of any cause, thus these models are not directly comparable. The relatively low number of patients further reduced the C-statistic in our study.

While our dataset does not allow a formal comparison with other biomarker models, the results of this study provide important biologic insights into the potential roles of angiogenic biomarkers of prognosis and disease recurrence in HCC patients undergoing LR or LT. Plasma VEGF and VEGF-C are candidate biomarkers that should be further validated to improve patient stratification.

## 4. Materials and Methods

### 4.1. Patients, Diagnosis and Therapeutic Strategy

The study enrolled consecutive patients diagnosed with HCC who underwent surgical interventions at the Center for Digestive Disease and Liver Transplantation of Fundeni Clinical Institute, Bucharest, Romania between 2003 and 2016. Samples were collected from all eligible patients who signed informed consent.

All HCC patients who were selected to undergo surgery were preoperatively evaluated by computed tomography (CT) or enhanced magnetic resonance imaging (MRI). Chest CT, bone scan and positron emission tomography (PET)/CT were performed to exclude distant metastases and other primary malignancies. We evaluated the blood levels of AFP as a standard clinical biomarker, as well as hepatitis viral markers B, C and D (HBV, HCV and HDV) and liver function parameters. CTP score and MELD were used for liver function evaluation prior to surgery [5].

At the time of HCC diagnosis, surgical treatment selection was based on patient’s liver function using the BCLC scoring system and the biologic behavior of the tumor. According to the BCLC treatment algorithm, curative options are limited to early stage HCC (BCLC stage 0 or A) and include LR, tumor ablation and LT [44]. Prior to patient selection LT or LR, we discussed the case with the multidisciplinary team consisting of hepatologists, medical oncologists, hepato-biliary surgeons and interventional radiologists. We followed institutional guidelines, according to which LR was the treatment of choice for early stage HCC in non-cirrhotic patients or in patients with cirrhosis, but well-preserved liver function. LT was recommended in HCC patients with cirrhosis and/or with increased portal pressure or bilirubin. HCC-related macrovascular invasion and extrahepatic tumor spread were considered contraindications for surgery. Moreover, only patients receiving curative resection were selected for this study. The curative resection was defined as complete removal of the tumors, with pathologically negative margins, either with LR or LT. In LT, complete tumor excision is achieved by total hepatectomy.

Eligible patients underwent LT taking into account the Milan criteria, determined by pretransplant radiological imaging. Pre-surgical locoregional treatments included TACE, radiofrequency ablation and percutaneous ethanol injection and were used for HCC down-staging for HCC patients on the waiting list for LT. The study conformed with the ethical guidelines of the 1975 Declaration of Helsinki and was approved by the Fundeni Clinical Institute Review Board (30,884/22.10.2014). No organs from executed prisoners were used for LT.

### 4.2. Study Design

All patients enrolled in this study were diagnosed with HCC. Liver tumor samples were collected at the time of surgery. Histopathological parameters and tumor staging at diagnosis were determined and combined with surgical records and perioperative imaging. Tumor grading was performed according to the Edmondson–Steiner classification.

OS was calculated as the period of time (in months) from the time of HCC resection until date of death or the last follow-up visit. Tumor recurrence was diagnosed according to EASL-criteria. The follow-up after HCC treatment with curative intent treatment included MRI or CT imaging and AFP determination at 3–4-month intervals for the first two years and extended to 6-month intervals thereafter. If recurrence was suspected, lesions were confirmed by contrast-enhanced MRI and lung CT. After confirmed recurrence, depending on the extent and localization of the tumors, patients were assessed and received surgical, TACE or radiofrequency ablation treatment. Patients with recurrence who were not eligible for these locoregional therapies were offered systemic therapy, including sorafenib and palliative radiotherapy for bone metastases. DFS was calculated as the period of time (in months) from the date of LT to the date of detection of the first tumor recurrence.

### 4.3. Measurement of Pro-Inflammatory and Angiogenic Biomarkers

Blood samples were obtained from all patients 1–2 days prior to surgery, processed for serum or plasma separation and aliquoting and stored at <−78 °C until analysis. These samples were used to measure circulating concentrations of pro-angiogenic and inflammatory biomarkers. The biomarkers evaluated included VEGF, placental growth factor (PlGF), VEGF-C, VEGF-D, sVEGFR1, basic fibroblast growth factor (bFGF) and sTIE-2 (using a commercially available 7-plex Growth Factor array, MesoScale Discovery, Gaithersburg, MD, USA); interferon (IFN)-γ, TNF-α and IL-1β, IL-2, IL-4, IL-6, IL-8, IL-10 and IL-12 heterodimer p70 (using a 9-plex Inflammatory Factor array, MesoScale Discovery). All samples were measured in duplicate on a commercially available MSD SECTOR Imager 2400 (MesoScale Discovery) in the CLIA-certified core of the Steele Laboratories at Massachusetts General Hospital, Boston, USA. AFP was measured as part of the standard clinical protocol.

To evaluate the known downstream effectors of IL10 pathway, we analyzed processed HCC TCGA data from cbioportal.org (TCGA-HCC-Provisional). Gene expression levels were centralized using a z-score of the median mRNA expression data for each gene separately, where a z-score > 2 was defined as “high”. For comparison, the data were log2-transformed. All editing, analysis and visualizations were performed using R software.

### 4.4. Statistical Analysis

We defined the outcome of HCC patients as a variable including any local tumor recurrence or distant-organ metastasis, denominated as DFS. Biomarker correlations with FLR, DFS and OS were tested using Wald test in univariate Cox regression analysis. Analysis was performed in proportional hazards regression model, with log-transformed biomarker levels and after stratifying based on Milan criteria, which included explant pathological data. Comparison of biomarker levels was performed using exact Wilcoxon test. Likelihood ratio test was performed to compare the model including only Milan criteria (as a stratum variable or in regression) and AFP, with a full model including Milan criteria, AFP and VEGF or VEGF-C.

## 5. Conclusions

In summary, examination of plasma biomarkers in surgical HCC patients identified differential correlations for angiogenic biomarkers (sVEGFR1, VEGF and VEGF-C) with outcomes after LT and for inflammatory biomarkers (particularly IL-10) with outcomes after LR. Moreover, we report that plasma VEGF and VEGF-C are candidate biomarkers for refining and expanding Milan criteria for LT. These hypothesis-generating data need to be independently validated in future studies.

## Figures and Tables

**Figure 1 cancers-12-01275-f001:**
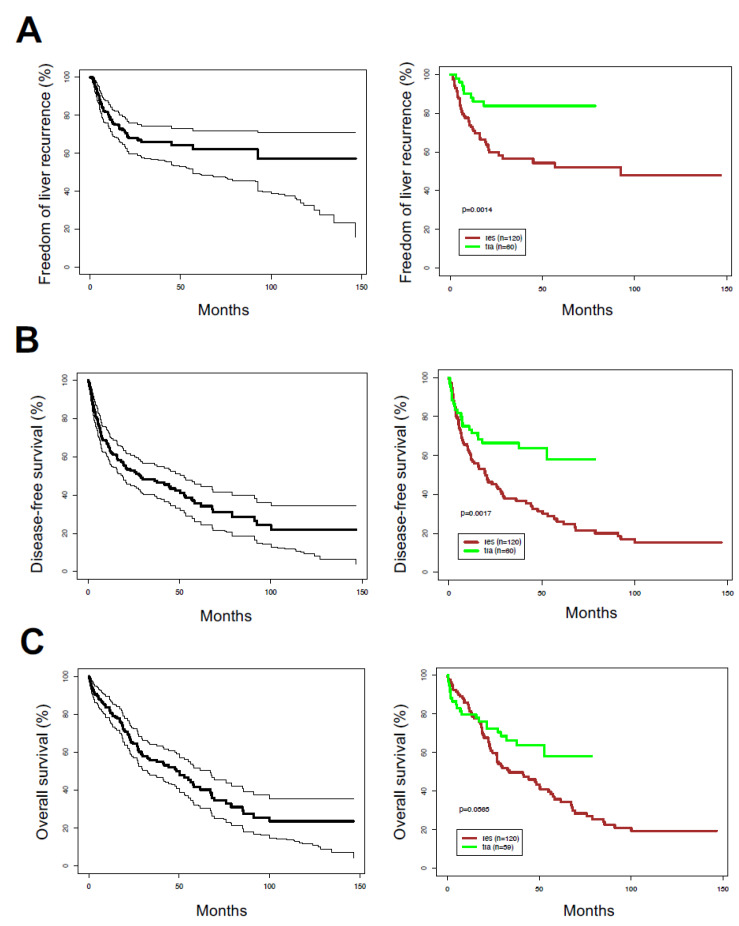
Kaplan–Meier survival distributions in hepatocellular carcinoma patient cohort. (**A**) Freedom of liver recurrence; (**B**) disease-free survival; (**C**) overall survival. p-values are derived from the log-rank test. Left plots show data for all patients, right graphs show data after segregating by surgical treatment: liver resection (res) versus transplantation (tra).

**Table 1 cancers-12-01275-t001:** Patient characteristics. Study enrolled 180 consecutive and eligible hepatocellular carcinoma patients. The *p* values are from the Wilcoxon/Fisher’s exact test.

Characteristics/ Surgical Intervention		Liver Resection	Liver Transplantation	*p*-Value
Age, years (*n*)		64 (16, 79) (120)	57 (30, 68) (60)	*<*0.0001
Milan criteria (*n* within, %)		45/119 (37.8%)	33/51 (64.7%)	0.0015
Edmondson Steiner grade (*n*, %)				0.044
I/II		10/116 (8.6%)	10/44 (22.7%)	
II/III		96/116 (82.8%)	31/44 (70.5%)	
III/IV		10/116 (8.6%)	3/44 (6.8%)	
Tumor size (cm ± SD, *n*)		7.0 ± 3.6 (118)	4.0 ± 1.7 (53)	*<*0.0001
Nodules (*n*, %)				0.0009
1		100/120 (83.3%)	35/57 (61.4%)	
2		14/120 (11.7%)	11/57 (19.3%)	
3		3/120 (2.5%)	8/57 (14.0%)	
4+		3/120 (2.5%)	3/57 (3.7%)	
Nodules (n ± SD, *n*)		1.2 ± 0.6 (120)	1.8 ± 1.4 (57)	0.0007
INR (range, *n*)		1.1 (1.0, 1.2) (41)	1.4 (1.2, 1.5) (54)	*<*0.0001
Pre-operative HBV (*n*, %)		49/119 (41.2%)	42/60 (70.0%)	0.0003
Pre-operative HCV (*n*, %)		47/119 (39.5%)	15/60 (25.0%)	0.067
Pre-operative HDV (*n*, %)		7/119 (5.9%)	26/60 (43.3%)	*<*0.0001
Pre-operative cirrhosis (*n*, %)		80/119 (67.2%)	59/60 (98.3%)	*<*0.0001
CTP score (*n*, %)	0	5/120 (29.2%)	0/59 (0.0%)	*<*0.0001
	A	78/120 (65.0%)	17/59 (28.8%)	
	B	7/120 (5.8%)	42/59 (71.2%)	
MELD score (median, range)		8 (6–11)	13 (7–28)	*<*0.0001
Surgery type (*n*, %)				*<*0.0001
Liver resection		108/120 (90.0%)	0/60 (0.0%)	
Liver resection + RFA/sorafenib/TACE/PCT		12/120 (10.0%)	0/60 (0.0%)	
Liver transplantation		0/120 (0.0%)	59/60 (98.3%)	
Liver transplantation + RFA/TACE		0/120 (0.0%)	1/60 (1.7%)	

CTP, Child–Turcotte–Pugh; HBV, hepatitis B virus; HCV, hepatitis C virus; HDV, hepatitis D virus; INR, international normalized ratio; RFA, radiofrequency ablation.

**Table 2 cancers-12-01275-t002:** Pre-operative plasma biomarkers in hepatocellular carcinoma patients who underwent liver resection (LR) and liver transplantation (LT). Data are shown as median values and interquartile ranges, with *p* values from the Wilcoxon/Fisher’s exact test.

Biomarker	LR Patients (*n*)	LT Patients (*n*)	*p* Value
IFN-γ (pg/mL)	7.5 (5.6, 13.3) (44)	7.8 (4.9, 13.4) (58)	0.91
IL-6 (pg/mL)	1.72 (1.60, 4.59) (44)	4.8 (2.0, 30.7) (58)	0.0001
IL-8 (pg/mL)	13.2 (7.4, 18.7) (44)	22.6 (11.3, 51.3) (58)	0.0017
IL-10 (pg/mL)	0.80 (0.57, 2.07) (45)	1.23 (0.59, 4.49) (58)	0.11
TNF-α (pg/mL)	3.5 (2.8, 4.1) (44)	3.4 (2.6, 5.0) (58)	0.84
bFGF (pg/mL)	4.3 (2.5, 8.6) (44)	12.1 (6.2, 32.6) (58)	<0.0001
PlGF (pg/mL)	34.6 (28.8, 43.0) (44)	27.2 (18.9, 35.1) (58)	0.0007
sVEGFR1 (pg/mL)	116 (87, 230) (44)	297 (134, 1166) (58)	0.0002
sTIE2 (pg/mL)	5334 (4610, 7429) (44)	6316 (5293, 8255) (58)	0.046
VEGF (pg/mL)	82 (42, 152) (44)	85 (39, 243) (58)	0.97
VEGF-C (pg/mL)	102 (89, 193) (44)	131 (89, 179) (58)	0.26
VEGF-D (pg/mL)	697 (549, 987) (44)	840 (681, 1162) (58)	0.012
AFP (ng/mL)	12.8 (6.3, 20.0) (31)	7.1 (3.3, 20.0) (53)	0.13

AFP, alpha-fetoprotein; bFGF, basic fibroblast growth factor; IFN, interferon; IL, interleukin; LR liver resection; LT, liver transplantation; PlGF, placental growth factor; sTIE2, soluble angiopoietin receptor; VEGF, vascular endothelial growth factor; VEGFR, VEGF receptor; TNF, tumor necrosis factor.

**Table 3 cancers-12-01275-t003:** Association between background clinical variables and freedom of liver recurrence (FLR), disease-free survival (DFS) and overall survival (OS) in hepatocellular carcinoma patients who underwent surgical treatments. *p*-Value from Wald test in a univariable Cox regression using log-transformed covariates.

Clinical Variables/ Outcome	FLR		DFS		OS	
	HR [95% CI] (*n*)	*p*-Value	HR [95% CI] (*n*)	*p*-Value	HR [95% CI] (*n*)	*p*-Value
All patients						
Milan criteria	0.58 [0.33,1.03] (173)	0.065	0.64 [0.43, 0.97] (173)	0.034	0.53 [0.34, 0.81] (173)	0.0037
Edmondson–Steiner grade	4.45 [0.97,20.40] (160)	0.054	1.75 [0.82, 3.71] (160)	0.15	1.41 [0.63, 3.16] (160)	0.40
Tumor size	1.03 [0.96, 1.11] (171)	0.38	1.06 [1.00, 1.12] (171)	0.050	1.06 [1.00, 1.12] (170)	0.042
HBV	2.08 [1.20, 3.63] (179)	0.0097	1.29 [0.87, 1.91] (179)	0.21	1.06 [0.70, 1.61] (178)	0.77
HCV	0.73 [0.41, 1.30] (179)	0.28	0.83 [0.55, 1.24] (179)	0.36	0.90 [0.59, 1.38] (178)	0.63
HDV	1.48 [0.63, 3.48] (179)	0.37	1.03 [0.54, 1.98] (179)	0.92	0.85 [0.44, 1.65] (178)	0.64
LT						
Milan criteria	0.38 [0.09, 1.51] (54)	0.17	0.48 [0.20, 1.16] (54)	0.10	0.53 [0.21,1.31] (54)	0.17
Edmondson–Steiner grade	NA [NA, NA] (44)	N/A	1.48 [0.27, 8.11] (44)	0.65	1.58 [0.29, 8.66] (44)	0.60
Tumor size	1.54 [0.97, 2.45] (53)	0.067	1.27 [0.96, 1.66] (53)	0.067	1.26 [0.95, 1.67] (52)	0.11
HBV	1.69 [0.34, 8.39] (60)	0.52	1.33 [0.52, 3.41] (60)	0.55	1.45 [0.53, 3.98] (59)	0.47
HCV	0.78 [0.16, 3.89] (60)	0.77	0.56 [0.19, 1.65] (60)	0.29	0.45 [0.13, 1.52] (59)	0.20
HDV	0.73 [0.17, 3.05] (60)	0.66	0.69 [0.29, 1.65] (60)	0.41	0.71 [0.29, 1.71] (59)	0.45
LR						
CTP Score	2.72 [1.05, 7.05] (120)	0.039	2.71 [1.41, 5.23] (120)	0.0029	3.26 [1.66, 6.42] (120)	0.0006
Milan criteria	0.64 [0.34, 1.18] (119)	0.15	0.70 [0.44, 1.09] (119)	0.11	0.53 [0.32, 0.86] (119)	0.011
Edmondson–Steiner grade	3.79 [0.83, 17.31] (116)	0.086	1.82 [0.76, 4.35] (116)	0.18	1.39 [0.55, 3.52] (116)	0.49
Tumor size	1.02 [0.95, 1.10] (118)	0.067	1.05 [0.99, 1.11] (118)	0.10	1.05 [0.99, 1.12] (118)	0.083
HBV	2.14 [1.19, 3.86] (119)	0.011	1.28 [0.83, 1.98] (119)	0.27	0.00 [0.62, 1.57] (119)	0.97
HCV	0.72 [0.39, 1.33] (119)	0.30	0.89 [0.57, 1.39] (119)	0.61	1.03 [0.64, 1.64] (119)	0.91
HDV	2.13 [0.84, 5.42] (119)	0.11	1.66 [0.72, 3.84] (119)	0.24	1.07 [0.43, 2.65] (119)	0.89

CTP, Child–Turcotte–Pugh; HBV, hepatitis B virus; HCV, hepatitis C virus; HDV, hepatitis D virus; LR liver resection; LT, liver transplantation.

**Table 4 cancers-12-01275-t004:** Association between selected plasma biomarkers and freedom of liver recurrence (FLR), disease-free survival (DFS) and overall survival (OS) in hepatocellular carcinoma patients who underwent surgical treatments. *p*-Value from Wald test in a univariable Cox regression using log-transformed covariates.

Biomarkers/ Outcome	FLR		DFS		OS	
	HR (95% CI) (*n*)	*p*-Value	HR (95% CI) (*n*)	*p*-Value	HR (95% CI) (*n*)	*p*-Value
All patients						
sVEGFR1	0.88 (0.65,1.18) (102)	0.40	0.94 (0.78,1.13) (102)	0.50	0.90 (0.72,1.13) (101)	0.38
VEGF	1.48 (1.09, 2.01) (102)	0.012	1.25 (1.04, 1.51) (102)	0.019	1.13 (0.92, 1.39) (101)	0.25
VEGF-C	1.31 (0.74, 2.32) (102)	0.35	1.49 (0.97, 2.29) (102)	0.071	1.67 (0.98, 2.85) (101)	0.058
IFN-γ	0.78 (0.51, 1.19) (102)	0.25	0.74 (0.55, 1.01) (102)	0.056	0.77 (0.55, 1.08) (101)	0.13
IL-6	0.99 (0.80, 1.24) (102)	0.95	1.02 (0.89, 1.17) (102)	0.72	1.01 (0.87, 1.17) (101)	0.87
IL-8	1.18 (0.89, 1.57) (102)	0.25	1.12 (0.93, 1.34) (102)	0.24	1.04 (0.85, 1.27) (101)	0.70
IL-10	0.88 (0.67, 1.17) (102)	0.39	1.01 (0.87, 1.17) (102)	0.93	1.01 (0.86, 1.17) (101)	0.95
AFP	1.04 (0.69, 1.56) (84)	0.85	0.96 (0.75, 1.24) (84)	0.77	0.97 (0.74, 1.28) (83)	0.84
LT						
sVEGFR1	0.69 (0.41, 1.14) (58)	0.15	0.78 (0.60, 1.02) (58)	0.069	0.74 (0.55, 0.99) (57)	0.045
VEGF	1.51 (0.96, 2.38) (58)	0.073	1.38 (1.07, 1.77) (58)	0.012	1.31 (1.02, 1.69) (57)	0.037
VEGF-C	2.19 (0.51, 9.51) (58)	0.29	2.47 (1.08, 5.64) (58)	0.033	2.49 (1.05, 5.93) (57)	0.039
IFN-γ	0.68 (0.32, 1.43) (58)	0.31	0.66 (0.42, 1.04) (58)	0.071	0.61 (0.38, 0.99) (57)	0.043
IL-6	0.93 (0.67, 1.27) (58)	0.63	0.97 (0.82, 1.15) (58)	0.71	0.92 (0.74, 1.21) (57)	0.39
IL-8	0.92 (0.61, 1.38) (58)	0.68	0.98 (0.78, 1.23) (58)	0.84	0.95 (0.61, 1.38) (57)	0.65
IL-10	0.88 (0.61, 1.25) (58)	0.47	0.92 (0.76, 1.11) (58)	0.37	0.86 (0.69, 1.07) (57)	0.18
AFP	0.89 (0.48, 1.65) (53)	0.71	0.89 (0.63, 1.25) (53)	0.50	0.93 (0.65, 1.33) (52)	0.69
LR						
sVEGFR1	1.03 (0.75, 1.42) (44)	0.84	1.15 (0.92, 1.42) (44)	0.22	1.38 (0.99, 1.92) (44)	0.061
VEGF	1.44 (0.96, 2.20) (44)	0.078	1.08 (0.81, 1.44) (44)	0.61	0.76 (0.53, 1.09) (44)	0.14
VEGF-C	1.18 (0.63, 2.23) (44)	0.60	1.21 (0.72, 2.04) (44)	0.47	1.30 (0.66, 2.56) (44)	0.45
IFN-γ	0.84 (0.50, 1.41) (44)	0.51	0.83 (0.55, 1.27) (44)	0.39	1.06 (0.63, 1.80) (44)	0.81
IL-6	1.08 (0.78, 1.51) (44)	0.63	1.17 (0.92, 1.50) (44)	0.20	1.34 (1.03, 1.74) (44)	0.032
IL-8	1.63 (1.09, 2.45) (44)	0.018	1.54 (1.11, 2.12) (44)	0.0093	1.35 (0.93, 1.96) (44)	0.12
IL-10	0.89 (0.56, 1.42) (44)	0.63	1.35 (1.01, 1.79) (44)	0.040	1.78 (1.28, 2.49) (44)	0.0007
AFP	1.17 (0.67, 2.05) (31)	0.59	1.06 (0.72, 1.58) (31)	0.50	1.04 (0.66, 1.63) (31)	0.87

AFP, alpha-fetoprotein; IFN, interferon; IL, interleukin; LR liver resection; LT, liver transplantation; VEGF, vascular endothelial growth factor; VEGFR, VEGF receptor.

**Table 5 cancers-12-01275-t005:** Association between selected plasma biomarkers and freedom of liver recurrence (FLR), disease-free survival (DFS) and overall survival (OS) in hepatocellular carcinoma patients within Milan criteria who underwent surgical treatments. *p*-Value from Wald test in a univariable Cox regression using log-transformed covariates.

Biomarkers/ Outcome	FLR		DFS		OS	
	HR (95% CI) (*n*)	*p*-Value	HR (95% CI) (*n*)	*p*-Value	HR (95% CI) (*n*)	*p*-Value
LT						
sVEGFR1	0.49 (0.20, 1.17) (35)	0.11	0.64 (0.41, 0.99) (35)	0.044	0.63 (0.40, 0.98) (35)	0.042
VEGF	1.45 (0.83, 2.51) (35)	0.19	1.45 (1.04, 2.02) (35)	0.027	1.44 (1.03, 2.01) (35)	0.031
VEGF-C	2.12 (0.28, 16.23) (35)	0.47	1.85 (0.56, 6.06) (35)	0.31	2.02 (0.60, 6.79) (35)	0.26
IFN-γ	0.84 (0.33, 2.14) (35)	0.72	0.71 (0.38, 1.33) (35)	0.28	0.71 (0.38, 1.34) (35)	0.29
IL-6	0.85 (0.52, 1.38) (35)	0.50	0.89 (0.68, 1.17) (35)	0.41	0.89 (0.68, 1.17) (35)	0.40
IL-8	0.83 (0.46, 1.50) (35)	0.54	0.94 (0.68, 1.30) (35)	0.73	0.95 (0.69, 1.31) (35)	0.73
IL-10	0.79 (0.43, 1.45) (35)	0.45	0.91 (0.69, 1.20) (35)	0.51	0.91 (0.68, 1.20) (35)	0.49
AFP	0.97 (0.44, 2.15) (34)	0.94	1.00 (0.62, 1.61) (34)	0.99	0.98 (0.61, 1.58) (34)	0.95
LR						
sVEGFR1	3.08 (1.18, 8.08) (19)	0.022	2.40 (1.10, 5.26) (19)	0.028	1.52 (0.60, 3.85) (19)	0.38
VEGF	1.65 (0.92, 2.97) (19)	0.095	1.38 (0.83, 2.28) (19)	0.21	1.15 (0.47, 2.83) (19)	0.76
VEGF-C	1.75 (0.48, 6.41) (19)	0.40	2.19 (0.65, 7.35) (19)	0.20	2.45 (0.43, 13.90) (19)	0.31
IFN-γ	0.63 (0.22, 1.75) (19)	0.37	0.48 (0.20, 1.16) (19)	0.10	0.20 (0.02, 1.89) (19)	0.16
IL-6	0.58 (0.18, 1.86) (19)	0.36	1.05 (0.63, 1.75) (19)	0.85	1.67 (0.82, 3.39) (19)	0.15
IL-8	1.57 (0.84, 2.95) (19)	0.16	1.45 (0.88, 2.39) (19)	0.14	1.50 (0.79, 2.83) (19)	0.22
IL-10	0.49 (0.15, 1.57) (19)	0.23	1.16 (0.63, 2.14) (19)	0.63	1.94 (0.88, 4.29) (19)	0.10
AFP	1.39 (0.64, 3.00) (16)	0.40	1.07 (0.65, 1.77) (16)	0.78	0.83 (0.42, 1.62) (16)	0.58

AFP, alpha-fetoprotein; IFN, interferon; IL, interleukin; LR liver resection; LT, liver transplantation; VEGF, vascular endothelial growth factor; VEGFR, VEGF receptor.

**Table 6 cancers-12-01275-t006:** Association between selected plasma biomarkers and freedom of liver recurrence (FLR), disease-free survival (DFS) and overall survival (OS) in hepatocellular carcinoma patients outside Milan criteria who underwent surgical treatments. *p*-value from Wald test in a univariable Cox regression using log-transformed covariates.

Biomarkers/ Outcome	FLR		DFS		OS	
	HR (95% CI) (*n*)	*p*-Value	HR (95% CI) (*n*)	*p*-Value	HR (95% CI) (*n*)	*p*-Value
LT						
sVEGFR1	1.00 (0.54, 1.83) (17)	0.99	1.00 (0.68, 1.47) (17)	0.99	0.89 (0.57, 1.41) (17)	0.62
VEGF	2.00 (0.86, 4.65) (17)	0.11	1.53 (0.94, 2.49) (17)	0.085	1.34 (0.81, 2.23) (17)	0.25
VEGF-C	127.32 (0.20, 80,654) (17)	0.14	6.15 (1.60, 23.62) (17)	0.0082	5.41 (1.42, 20.64) (17)	0.014
IFN-γ	0.48 (0.15, 1.55) (17)	0.22	0.75 (0.37, 1.50) (17)	0.42	0.72 (0.34, 1.54) (17)	0.40
IL-6	1.07 (0.73, 1.59) (17)	0.72	1.14 (0.91, 1.44) (17)	0.26	1.06 (0.82,1.38) (17)	0.65
IL-8	1.31 (0.53, 3.24) (17)	0.57	1.36 (0.79, 2.34) (17)	0.26	1.30 (0.73, 2.29) (17)	0.37
IL-10	0.98 (0.61, 1.58) (17)	0.93	1.02 (0.77, 1.34) (17)	0.91	0.91 (0.64, 1.28) (17)	0.58
AFP	0.71 (0.21, 2.36) (13)	0.57	1.18 (0.57, 2.45) (13)	0.65	1.60 (0.67, 3.83) (13)	0.29
LR						
sVEGFR1	0.80 (0.45, 1.42) (25)	0.44	1.06 (0.80, 1.39) (25)	0.69	1.43 (0.97, 2.10) (25)	0.073
VEGF	1.26 (0.74, 2.14) (25)	0.39	0.97 (0.71, 1.32) (25)	0.84	0.78 (0.52, 1.16) (25)	0.22
VEGF-C	1.03 (0.47, 2.27) (25)	0.94	1.03 (0.56, 1.90) (25)	0.92	1.19 (0.60, 2.35) (25)	0.61
IFN-γ	0.93 (0.52, 1.65) (25)	0.81	0.99 (0.61, 1.59) (25)	0.96	1.26 (0.73, 2.16) (25)	0.40
IL-6	1.33 (0.91, 1.94) (25)	0.15	1.23 (0.91, 1.66) (25)	0.17	1.13 (0.80, 1.58) (25)	0.49
IL-8	1.59 (0.93, 2.71) (25)	0.093	1.53 (0.99, 2.36) (25)	0.054	1.35 (0.78, 2.34) (25)	0.28
IL-10	1.09 (0.67, 1.78) (25)	0.79	1.38 (0.99, 1.92) (25)	0.060	1.56 (1.07, 2.27) (25)	0.021
AFP	0.82 (0.37, 1.82) (15)	0.63	1.05 (0.54, 2.06) (15)	0.88	1.24 (0.66, 2.32) (15)	0.50

AFP, alpha-fetoprotein; IFN, interferon; IL, interleukin; LR liver resection; LT, liver transplantation; VEGF, vascular endothelial growth factor; VEGFR, VEGF receptor.

**Table 7 cancers-12-01275-t007:** Analysis of plasma VEGF and VEGF-C combined with Milan criteria and AFP in hepatocellular carcinoma patients who underwent liver transplantation. Analysis was performed in proportional hazards regression model, with log-transformed VEGF or VEGF-C and AFP measurements and either stratifying on Milan criteria as a stratum variable or including Milan criteria in regression. Data are presented as C-statistic with 95% confidence intervals.

Biomarkers/ Outcome	Plasma VEGF			Plasma VEGF-C		
DFS	C-statistic	χ^2^	*p*-value	C-statistic	χ^2^	*p*-value
Milan as stratum	0.690 (0.578, 0.802)	8.22	0.0042	0.620 (0.453, 0.788)	6.18	0.013
Milan in regression	0.729 (0.624, 0.835)	8.44	0.0037	0.692 (0.555, 0.830)	5.95	0.015
OS	C-statistic	χ^2^	*p*-value	C-statistic	χ^2^	*p*-value
Milan as stratum	0.674 (0.562, 0.785)	6.12	0.013	0.629 (0.461, 0.797)	5.86	0.016
Milan in regression	0.700 (0.585, 0.815)	5.98	0.015	0.686 (0.543, 0.830)	6.54	0.011

AFP, alpha-fetoprotein; VEGF, vascular endothelial growth factor.

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
