# Peer review of "Potential Circulating Biomarkers of Recurrence after Hepatic Resection or Liver Transplantation in Hepatocellular Carcinoma Patients"

_cancers, 2020, doi:10.3390/cancers12051275_

Round 1
Reviewer 1 Report
Thank you so much for giving me a precious opportunity to review this revised manuscript.
I checked your revised contents. It was acceptable for me and I really appreciate these author's efforts.
Author Response
We thank the reviewer for his/her evaluation and suggestions. As suggested, we have extensively edited the manuscript to improve clarity and grammar.
Reviewer 2 Report
Potential Circulating Biomarkers of Recurrence after Hepatic Resection or Liver Transplantation in Hepatocellular Carcinoma Patients
The authors improved the original article significantly. The major concerns were sufficiently answered. The shown data now offers a better insight into the trial and the multifactorial interaction of angiogenesis and inflammation in advanced cirrhosis and hepatocellular carcinoma. The clinical demand for additional biomarkers to extend the restrictive Milan-criteria with an acceptable overall survival and freedom of tumor recurrence is high.
Some minor revision needs to be performed.
The confidence interval in the C-statistic are missing. Thus, the significance can`t be sufficiently interpreted.
Author Response
The authors improved the original article significantly. The major concerns were sufficiently answered. The shown data now offers a better insight into the trial and the multifactorial interaction of angiogenesis and inflammation in advanced cirrhosis and hepatocellular carcinoma. The clinical demand for additional biomarkers to extend the restrictive Milan-criteria with an acceptable overall survival and freedom of tumor recurrence is high.
We thank the reviewer for his/her evaluation.
Some minor revision needs to be performed.
We agree and have further edited the manuscript to improve clarity and grammar.
The confidence interval in the C-statistic are missing. Thus, the significance can`t be sufficiently interpreted.
We agree and have included the confidence intervals for the C-statistic values in the revised Table 7.
This manuscript is a resubmission of an earlier submission. The following is a list of the peer review reports and author responses from that submission.
Round 1
Reviewer 1 Report
This manuscript was interesting and their conclusion would influence future research, however I have some comments.
- How did you decide liver resection or liver transplantation? The severity of liver damage (MELD score or Child-Pugh score) would influence outcome after surgery. The timing of liver transplantation for HCC especially was challenging irrespective of Milan criteria.
- It was a little difficult to understand their background, so why did they measure these biomarkers (sVEGFR1, VEGF and VEGF-C) and when did they measure these markers before surgery?
- How did they decide sample size?
- How did they diagnose recurrence of HCC? Did they follow the patients regularly?
Reviewer 2 Report
Potential Biomarkers of Recurrence after Hepatic Resection or Liver Transplantation in Hepatocellular Carcinoma Patients
The authors described the impact of plasma biomarkers in patients with hepatocellular carcinoma (HCC) undergoing liver transplantation (LT) and resection (LR). A special focus was on angiogenic in LT and inflammatory biomarkers in LR. Objective biomarkers guiding HCC treatment are needed in clinical routine. At present restrictive treatment algorithms commonly assign patients at first diagnosis to palliation while tailored treatment concepts might achieve superior results.
Even though the authors address an unsolved clinical demand; this study exhibits methodic weakness. Additionally, VEGF as a biomarker for overall survival and tumor recurrence after LT has been investigated in single center trials.
Lack of strength
- The study prospectively collected samples from patients undergoing surgical HCC treatment. The analysis was performed in a retrospective fashion.
- The process of selecting the study population is missing.
- The algorithm of assigning patients either to resection or liver transplantation and the policy of organ allocation and distribution are insufficiently described. It is not evident if the treatment stratification complies with international standards.
- Essential markers of cirrhosis, tumor invasiveness and treatment results are missing.
A major revision of this study is necessary.
Methods
- The authors must clarify the treatment algorithms and the process of selecting the study population.
- The authors need to comment whether all patients received curative resection, defined as complete resection of all microscopic and macroscopic tumors and exclusion of ruptured HCCs. Only patients with a complete resection should be considered eligible.
- Vascular infiltration needs to be determined in pre-treatment imaging and in histopathologic assessment of the resected tumor and liver tissue.
- The documentation of the extend of pre-operative liver insufficiency should be registered.
- The surveillance for HCC recurrence appears to be insufficient (AASLD guideline recommends abdominal and chest CT in transplanted patients).
Statistics
- Differences in liver function between the LT and LR cohorts should be documented (MELD, CHILD or ALBI)
- Biomarker correlation should be reperformed after assessment of suggested changes in Methods 2. /3. / 4. and stratification of the study population respectively.
- The model including Milan criteria, AFP and VEGF or VEGF-C should be compared for example in a ROC-analysis with other biomarker supported models. Benchmarks are the Metroticket project by ELITA (European Liver and Intestine Transplant Association) and ILTS (International Liver Transplantation Society) and the Duvoux criteria.
Discussion
- The discussion should further elucidate the impact of the extend of liver insufficiency on the assessed biomarkers. Table 2 offers limited insight into the liver function but is suggestive for a significant difference between the LT and LR cohort.
- In case the demanded modification in methods and statistics offer new options of interpretation the present discussion should be reconsidered.
